evolution, ecology, health and disease and epidemiology

host–parasite interaction, *Gasterosteus aculeatus*, *Schistocephalus solidus*, resistance, tolerance

**Author for correspondence:**
Agnes Piecyk
e-mail: apiecyk@evolbio.mpg.de

†Deceased. We dedicate this paper to the memory of Martin Kalbe.

# Cross-continental experimental infections reveal distinct defence mechanisms in populations of the three-spined stickleback *Gasterosteus aculeatus*

Agnes Piecyk[1,2], Megan A. Hahn[3], Olivia Roth[2,4], Nolwenn M. Dheilly[3], David C. Heins[5], Michael A. Bell[6] and Martin Kalbe[1,†]

[1]Evolutionary Ecology, Max Planck Institute for Evolutionary Biology, Plön, Germany
[2]Evolutionary Ecology of Marine Fishes, GEOMAR Helmholtz Centre for Ocean Research Kiel, Kiel, Germany
[3]School of Marine and Atmospheric Sciences, Stony Brook University, Stony Brook, NY, USA
[4]Marine Evolutionary Biology, Kiel University, Kiel, Germany
[5]Department of Ecology and Evolutionary Biology, Tulane University, New Orleans, LA, USA
[6]University of California Museum of Paleontology, University of California, Berkeley, CA, USA

AP, 0000-0002-9582-121X

Epidemiological traits of host–parasite associations depend on the effects of the host, the parasite and their interaction. Parasites evolve mechanisms to infect and exploit their hosts, whereas hosts evolve mechanisms to prevent infection and limit detrimental effects. The reasons why and how these traits differ across populations still remain unclear. Using experimental cross-infection of three-spined stickleback *Gasterosteus aculeatus* and their species-specific cestode parasites *Schistocephalus solidus* from Alaskan and European populations, we disentangled host, parasite and interaction effects on epidemiological traits at different geographical scales. We hypothesized that host and parasite main effects would dominate both within and across continents, although interaction effects would show geographical variation of natural selection within and across continents. We found that mechanisms preventing infection (qualitative resistance) occurred only in a combination of hosts and parasites from different continents, while mechanisms limiting parasite burden (quantitative resistance) and reducing detrimental effects of infection (tolerance) were host-population specific. We conclude that evolution favours distinct defence mechanisms on different geographical scales and that it is important to distinguish concepts of qualitative resistance, quantitative resistance and tolerance in studies of macroparasite infections.

## 1. Background

Epidemiological traits characterize the interaction and distribution of hosts and parasites and are shaped through the effects of the host, the parasite, their interaction and their environment [1–4]. Although host and parasite genotypes and allele frequencies change over evolutionary timescales, the response of an individual to different environmental conditions (known as 'reaction norm') is plastic. Understanding evolutionary dynamics and variation in host and parasite genetic and plastic effects on infection outcomes is crucial in basic science and clinical settings [5,6].

Parasites rely on host resources and evolve mechanisms increasing their ability to infect and to exploit their hosts [7]. Natural selection favours parasite traits that increase their fitness through trade-offs involving infectivity, growth and transmission. In response, hosts evolve defence mechanisms to resist and to tolerate parasites. Resistance reduces the likelihood of infection (qualitative

resistance) or limits parasite replication or growth (quantitative resistance), whereas tolerance limits the negative effects of a given parasite burden without reducing parasite replication or growth [4,8–11]. Resistance and tolerance are not mutually exclusive but have different ecological and evolutionary consequences [8,12,13]. For example, parasite prevalence is expected to decrease if hosts evolve resistance, whereas parasite prevalence may increase if hosts evolve tolerance [14,15]. The differences between resistance and tolerance have long been recognized in plant research but only recently came into focus for zoologists and clinicians [9,10,12,16]. Here, we show that the distinction between qualitative resistance, quantitative resistance and tolerance is of central importance in macroparasite infections. We focus on vertebrate defence mechanisms against helminths—parasitic worms that infect about two billion people worldwide, often establish long-lasting infections and cause substantial morbidity, mortality and economic loss [17–19]. We propose that suppression of the parasite's growth is a particularly important hitherto understudied form of quantitative resistance.

In this regard, the three-spined stickleback (*Gasterosteus aculeatus*; hereafter 'stickleback') and its species-specific cestode parasite *Schisocephalus solidus* provide an outstanding model to study host and parasite effects on epidemiological traits during the infection process [20–22]. Sticklebacks are distributed across the Northern Hemisphere where they have adapted to a wide range of habitats [23]. Stickleback populations differ in phenotypic and genotypic traits including morphology, behaviour and immunity. Local adaptation, divergent selection and genomic differentiation have been linked to abiotic factors, such as marine–freshwater divergence, and biotic factors, such as parasites [24–27]. Moreover, stickleback immune gene frequencies and levels of immunological activation differ among environments [28–30], with genetic adaptation and phenotypic plasticity contributing significantly to the stickleback's immune response [31–33].

Throughout their geographical range, freshwater stickleback are frequently infected by *S. solidus*. This trophically transmitted cestode penetrates the intestinal wall and enters the body cavity of the fish where it undergoes most of its somatic growth within weeks or months [22,34]. The relative weight of *S. solidus* in the fish, the parasite index (PI) [35], is a measure of parasite fitness [36,37], virulence [35,38,39] and host resistance [40]. The life cycle of *S. solidus* begins anew following transmission to the definitive host, mostly birds, when the eggs are defaecated into the water. The definitive host can be replaced by an *in vitro* breeding system, facilitating controlled experimental infections [41,42].

## (a) Approach and aim

To investigate genotypic and phenotypically plastic effects on different epidemiological traits, we used Alaskan and European hosts and parasites from geographically distant and adjacent populations in experimental cross-infections (figure 1a). We determined (i) the infection rates as a measure of parasite infectivity and host qualitative resistance, (ii) parasite size as a measure of virulence, transmission potential and host quantitative resistance, (iii) proxies of host body condition as measures of tolerance and costs of resistance and (iv) host immunological parameters including regulatory and immune gene expression as measures of the molecular host–parasite interplay. We hypothesized (i) baseline differences between host populations within and across continents (indicating host genotype effects) (ii) parasite–strain-specific responses to infection within and across continents (indicating parasite genotype effects and phenotypically plastic host responses) and (iii) different interaction effects at different geographical scales.

## 2. Results

We distinguish between qualitative resistance (infection success), quantitative resistance (parasite growth) and tolerance (strength of infection effects) to disentangle host, parasite and interaction effects on epidemiological traits during the infection process.

## (a) Qualitative resistance is combination specific whereas quantitative resistance is mainly determined by host effects

Host qualitative resistance and parasite infectivity were determined via *S. solidus* infection rates. The infection rates in copepods (first intermediate hosts) neither differed significantly between rounds nor between parasite populations (electronic supplementary material, SI.1). The infection rates in stickleback were significantly affected by an interaction between host and parasite populations (generalized linear mixed-effects models (GLMM); $p = 0.006$). Among continents, Alaskan *S. solidus* from both Wolf and Walby infected European DE stickleback, but European NO parasites failed to infect stickleback from Wolf. Infection rates of geographically adjacent populations were higher in sympatric than in allopatric combinations ($\chi^2_1 = 5.6504$; $p = 0.0175$).

PIs, approximations of host quantitative resistance, differed between, but not within host populations. All *S. solidus* strains grew largest in European DE stickleback (LMMs; each $p < 0.01$; electronic supplementary material, SI.2: figure 2). Whether the Alaskan combinations were sympatric or allopatric had no significant influence on the PI ($\chi^2_1 = 0.0283$; $p = 0.866$).

## (b) Body condition and immunological parameters differ between stickleback populations and in response to infection

Stickleback body condition was assessed through the condition factor (CF) and the hepatosomatic index (HSI) [43,44]. Fish condition (excepting the HSI between DE and Wolf) differed significantly between controls from the different populations. Wolf stickleback had the lowest condition, DE stickleback had the highest condition (electronic supplementary material, table S3 and figure S2). DE stickleback had larger head kidneys and spleens than stickleback from both Alaskan populations (GLMMs; each $p < 0.001$), but spleen size did not differ significantly between DE and Walby controls (electronic supplementary material, table S3 and figure S3). Differences between the populations remained if fish were exposed to *S. solidus* but uninfected (electronic supplementary material, table S4). The CF of DE stickleback and the HSI of Walby stickleback differed significantly between controls and exposed individuals, suggesting an

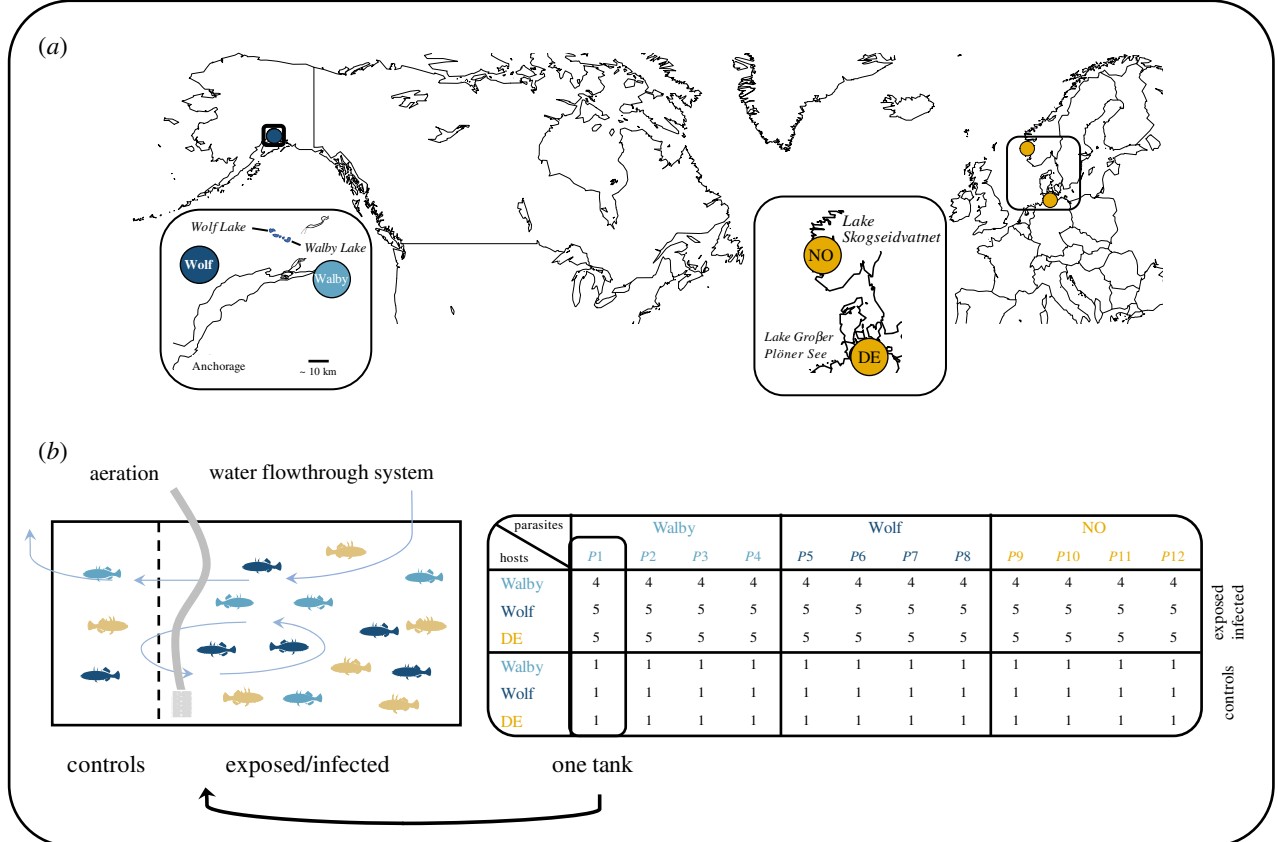

**Figure 1.** Sampling sites and experimental setup. (*a*) Stickleback (*Gasterosteus aculeatus*) and cestodes (*Schistocephalus solidus*) were sampled from Alaska (Walby and Wolf) and Europe (DE and NO). (*b*) The experiment was composed of three rounds with the same parasite sibships (*P*1 to *P*12) and different fish families. For each fish family, one individual fish was sham-exposed as a control. The table shows sample sizes from experimental rounds 2 and 3; round 1 included four exposed fish from Wolf. Fish from all populations (controls and *S. solidus* exposed and uninfected or infected) were housed in the same tanks. Light blue arrows illustrate the water current. Colours refer to the two Alaskan (Walby = light blue; Wolf = dark blue) and European populations (yellow = DE hosts or NO parasites). (Online version in colour.)

effect of parasite exposure (electronic supplementary material, tables S6 and S7).

Infection with Walby and NO *S. solidus* caused a significant decrease of the CF of DE hosts; Wolf infection was linked to a condition decrease in Walby and Wolf stickleback (electronic supplementary material, table S6). We detected a host-population-specific relation between host CF and infection intensity (PI) in Walby and Wolf infections (host population–PI interaction in Walby infections: $F_{2,21.7} = 9.37$, $p = 0.0012$; host population–PI interaction in Wolf infections: $F_{2,17.5} = 4.02$, $p = 0.037$). Although the CF decreased with increasing PI in Walby and DE fish, Wolf condition was not affected by PI (figure 3). HSIs were significantly lower in all infected fish, regardless of host and parasite origin and PI; effects did not differ between parasite origins within host populations (electronic supplementary material, table S7 and figure S2).

In each host population, *S. solidus* infection resulted in similar immunological parameters as inferred from the relative weight of the two major immune organs (splenosomatic index, SSI, head kidney index, HKI) (electronic supplementary material, table S5 and figure S3). The SSI generally increased upon infection. The effects were pronounced in specific combinations: Wolf and NO *S. solidus* in DE hosts, Walby and Wolf *S. solidus* in Wolf hosts, and Walby *S. solidus* in Walby hosts (electronic supplementary material, table S8 and figure S3). Head kidneys were larger in infected Alaskan stickleback; head kidneys of DE hosts were not significantly affected (electronic supplementary material, table S9 and figure S3).

## (c) Baseline regulatory and immune gene expression differs between stickleback populations

In order to test whether augmented sizes of the major immune organs indicated increased immunological activation in specific host–parasite combinations, total head kidney RNA was extracted from 84 controls, 101 exposed but uninfected fish (exposed) and 80 infected stickleback. We used a multivariate approach (i) grouping data from 25 targets (*total*) and (ii) grouping data according to functional groups: 11 innate immune genes (*innate*), eight adaptive immune genes (*adaptive*), three complement component genes (*complement*) and three regulatory genes (*regulatory*).

Stickleback population (PERMANOVA$_{total}$: $F_{2,264} = 5.96$, $p < 0.001$) and infection status (PERMANOVA$_{total}$: $F_{2,264} = 3.41$, $p < 0.001$) significantly affected *total* expression profiles; interactions were not significant (electronic supplementary material, table S10).

Gene expression profiles of controls differed between the Alaskan populations (PERMANOVA$_{total}$: $F_{1,52} = 2.60$, $p = 0.003$; PERMANOVA$_{complement}$: $F_{1,52} = 4.81$, $p = 0.007$) and between DE and Wolf stickleback (PERMANOVA$_{total}$: $F_{1,54} = 3.57$, $p = 0.007$; PERMANOVA$_{innate}$: $F_{1,54} = 2.72$, $p = 0.026$; PERMANOVA$_{complement}$: $F_{1,54} = 2.77$, $p = 0.023$; PERMANOVA$_{regulatory}$: $F_{1,54} = 5.77$, $p = 0.013$). *Total* expression profiles did not differ significantly between DE and Walby stickleback. However, multivariate analyses of functional groups indicated significantly different *regulatory* gene expression between DE and Walby controls

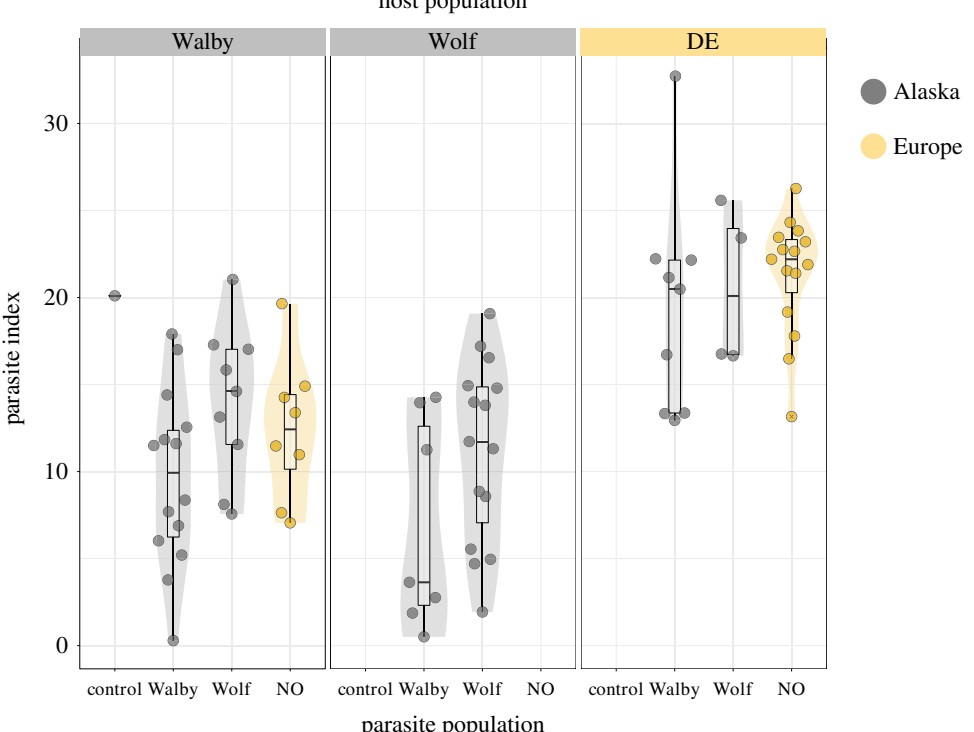

**Figure 2.** *Schistocephalus solidus* parasite indices in three different stickleback hosts. Alaskan (Walby and Wolf) and European (DE) stickleback were exposed to Alaskan (Walby and Wolf) and European (NO) *S. solidus*. The infection success and the parasite index (relative weight of the parasite in the host [35]) were determined after nine weeks. (Online version in colour.)

(PERMANOVA$_{regulatory}$: $F_{1,59} = 2.57$, $p = 0.012$) (electronic supplementary material, SI.4.1: figure S4 and tables S12–S14). Hierarchical clustering on Euclidean distances indicated the highest divergence of Wolf profiles (figure 4a). *A posteriori* analyses identified differential expression of seven out of 25 genes (figure 4a; electronic supplementary material, table S15).

### (d) Differences between stickleback populations remain if fish are exposed but converge upon infection

Host population differences remained if fish were exposed to *S. solidus* but uninfected (host effect: PERMANOVA$_{total}$: $F_{2,101} = 2.75$, $p = 0.0002$; parasite effect: PERMANOVA$_{total}$: $F_{2,101} = 0.5$, $p = 0.096$; host–parasite interaction: PERMANOVA$_{total}$: $F_{2,101} = 1.05$, $p = 0.428$). To understand the host effect in more detail, we used host population as explanatory and found gene expression profiles differing especially upon exposure to Wolf and NO *S. solidus* (Wolf exposure: PERMANOVA$_{total}$: $F_{2,31} = 2.1$, $p = 0.005$; PERMANOVA$_{adaptive}$: $F_{2,31} = 3.42$, $p < 0.001$; NO exposure: PERMANOVA$_{adaptive}$: $F_{2,36} = 4.75$, $p < 0.001$; electronic supplementary material, tables S16–S18). Gene expression profiles were not significantly affected by *S. solidus* strain within host populations (electronic supplementary material, SI 4.2: figure S5 and tables S16–S21).

Using LMMs to test which genes were differently expressed, we found that Wolf exposed Walby stickleback showed higher expression of five *adaptive* genes in comparison to Wolf or DE stickleback (figure 4b; electronic supplementary material, table S22). NO *S. solidus* exposed stickleback showed differential expression of four *adaptive* genes, of which three genes were more highly expressed in Walby than in Wolf (figure 4b; electronic supplementary material, table S23).

Focusing on infected individuals ($n = 80$), we found that gene expression profiles mostly converged upon infection (electronic supplementary material, SI.4.3: figure S6 and tables S24–S29). Only NO infection caused different *adaptive* gene expression profiles in Walby versus DE stickleback (PERMANOVA$_{adaptive}$: $F_{1,21} = 6.64$, $p < 0.001$; figure 4c; electronic supplementary material, tables S26 and S30).

### (e) Infection impacts gene expression in a parasite-dependent manner

We tested the effect of infection status (infected, exposed, control) on gene expression (electronic supplementary material, SI.4.4.) and ran pairwise comparisons of infected and control fish (electronic supplementary material, SI.4.5.), infected and exposed fish (electronic supplementary material, SI.4.6.), and control and exposed fish (electronic supplementary material, SI.4.7.). These analyses revealed a Wolf parasite effect on *innate* gene expression in Walby and Wolf hosts in comparison to controls (Wolf infection in Walby: PERMANOVA$_{innate}$: $F_{1,38} = 1.38$, $p = 0.009$; Wolf infection in Wolf: PERMANOVA$_{innate}$: $F_{1,38} = 1.57$, $p = 0.007$; electronic supplementary material, figures S12 and S13, tables S41 and S44). Walby *S. solidus* infection was associated with upregulation of *total*, *innate* and *regulatory* genes of DE stickleback compared to controls (PERMANOVA$_{total}$: $F_{1,38} = 5.71$, $p = 0.02$; PERMANOVA$_{innate}$: $F_{1,38} = 9.92$, $p = 0.004$; PERMANOVA$_{regulatory}$: $F_{1,38} = 7.12$, $p = 0.009$; electronic supplementary material, figure S14 and tables S45–S47). *Total*, *innate*, *adaptive* and *regulatory* profiles differed between Walby exposed and Walby infected DE stickleback (PERMANOVA$_{total}$: $F_{1,21} = 5.8$, $p = 0.007$; PERMANOVA$_{innate}$: $F_{1,21} = 8.85$, $p = 0.003$; PERMANOVA$_{adaptive}$: $F_{1,21} = 5.16$, $p = 0.006$; PERMANOVA$_{regulatory}$: $F_{1,21} = 7.43$, $p = 0.02$; electronic supplementary material, figure S15 and table S53). We further

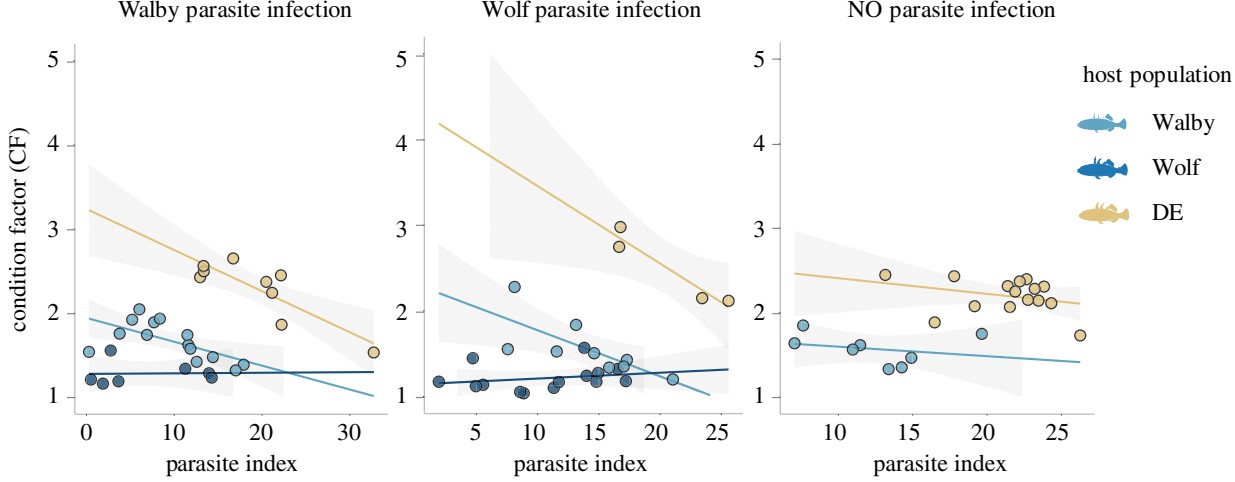

**Figure 3.** Host population-specific relation between body condition and infection intensity (i.e. tolerance). Each dot represents one individual; lines and shaded areas show linear regression fits with 95% confidence intervals. Colours indicate the host population. (Online version in colour.)

detected significant differences between NO exposed and NO infected DE stickleback (PERMANOVA$_{total}$: $F_{1,26} = 2.54$, $p = 0.02$; PERMANOVA$_{innate}$: $F_{1,61} = 5.12$, $p < 0.001$; PERMANOVA$_{adaptive}$: $F_{1,26} = 4.33$, $p < 0.001$; electronic supplementary material, table S55). Whether Alaskan stickleback were exposed or infected with sympatric or allopatric *S. solidus* did not affect gene expression profiles (electronic supplementary material, SI.4.7: tables S65 and S66).

## 3. Discussion

Host defence strategies are classified into mechanisms of resistance and tolerance. While resistance mechanisms reduce parasite burden by preventing infection (qualitative resistance) or by limiting parasite growth (quantitative resistance), tolerance mechanisms limit detrimental effects of a given parasite burden [4,8,11,16]. Here, we applied the concepts of qualitative resistance, quantitative resistance and tolerance on helminth infections of stickleback and determined effects of (i) the host, (ii) the parasite and (iii) host–parasite interactions on each of these epidemiological traits.

Our first key finding was that resistance and tolerance differed among host populations, implying host genetic effects on infection outcome. Parasite infection rates (i.e. host qualitative resistance) depended on host genotype–parasite genotype interaction, whereas parasite size (i.e. host quantitative resistance) was affected by the host but neither differed among parasite strains within host populations nor according to interaction effects. Our second key finding was that constitutive differences of gene expression profiles and immunological parameters among host populations remained upon parasite exposure but mostly converged upon infection. This result implies dominant effects of parasite-induced phenotypic plasticity and a stronger parasite genotype main effect compared to interaction effects.

### (a) Variation in host defence mechanisms

We observed two distinct types of resistance in combinations of geographically disparate populations of hosts and parasites. First, Wolf stickleback prevented infection by European *S. solidus* and sympatric Alaskan combinations yielded higher infection rates than allopatric combinations.

Second, stickleback from both Alaskan populations showed higher quantitative resistance than European stickleback.

Combination specific qualitative resistance against NO *S. solidus* has been reported before: stickleback from two out of three Canadian populations resisted NO *S. solidus* infection [45]. Whether inter-continental resistance can be attributed to local adaptation or specificities of host and parasite populations or clades, warrants further investigation that takes the effect of environmental variation on defence mechanisms and infection outcomes into consideration.

Quantitative resistance, i.e. the ability to control parasite growth, was approximated by the PI. Neither parasite origin, nor sympatry had an effect on parasite size and the PI did not differ significantly between Alaskan populations. In line with previous results [31,46], quantitative resistance was much lower in DE stickleback, indicating a dominant host effect.

The relationship of parasite size and host condition was used to estimate tolerance. In addition to the qualitative resistance of Wolf stickleback, these fish also appeared to be more tolerant than Walby and DE hosts (figure 3). Accordingly, stickleback populations (here, Wolf) can have both higher qualitative resistance and tolerance compared to stickleback from other, even nearby, populations. We suggest that high tolerance is a universal property of Wolf fish, whereas the prevention of infection is specific to NO *S. solidus*. Notably, *S. solidus* size depends on the size of the stickleback [47,48], and *vice versa* and the relative contribution of environmentally mediated phenotypic plasticity to infection phenotypes can be substantial [49,50]. Accordingly, what manifests as 'tolerance' could result from low host condition, which causes low parasite growth. Furthermore, the lack of an ecological context in laboratory experiments could obscure our results.

Immune defence is costly and might be selected against [51,52]. Our study confirms this assumption by demonstrating significantly lower body conditions in exposed than in control fish. In line with a previous study [29], exposure had no significant effect on gene expression. We cannot conclude whether exposed but ultimately uninfected stickleback had prevented or cleared infection. The parasites of uninfected fish may have failed to target and/or overcome the intestinal wall or were eliminated by the host's immune system.

Proc. R. Soc. B 288: 20211758

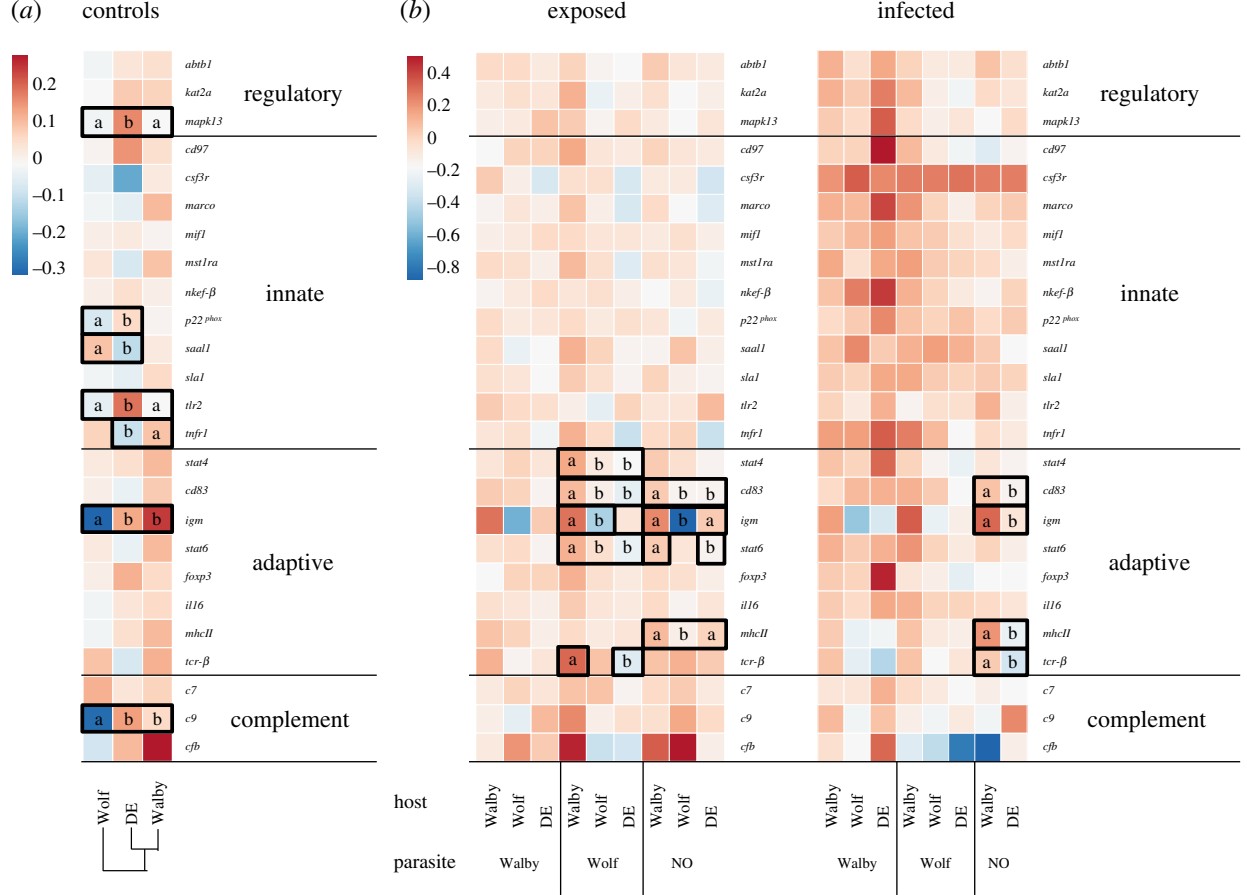

**Figure 4.** Stickleback gene expression profiles. Alaskan (Walby; Wolf) and European (DE) stickleback were sham-exposed (controls) or exposed to Alaskan (Walby; Wolf) or European (NO) *S. solidus*. Total RNA was extracted from head kidneys after nine weeks. The heatmaps are based on $\log_{10}$ transformed CNRQ values that were averaged across treatments. Lower case letters indicate significantly different expression of single genes. (a) Gene expression profiles of sham-exposed controls. The columns were hierarchically clustered on Euclidian distances. (b) Gene expression profiles of exposed but uninfected (exposed) and infected sticklebacks. Columns were ordered according to treatment. (Online version in colour.)

## (b) Parasite-induced phenotypic plasticity transcends host genetic differences

Our data indicate constitutive differences, thus genetic divergence, among stickleback populations: compared to Alaskan populations, DE stickleback were in better condition and had a higher baseline immunological activation. Gene expression profiles differed among all host populations. Especially Wolf stickleback profiles were distinct from DE and Walby (figure 4). Genetic divergence between European and North American stickleback and *S. solidus* is well documented [25,53–55]. We show that proxies of immunological activation converged upon infection, which has previously been shown for *S. solidus* infections of different European stickleback [31]. Although the precise molecular mechanisms will have to be investigated, our results suggest that *S. solidus* affects stickleback gene expression in a host–parasite genotype-dependent manner and that parasite-induced phenotypic plasticity transcends host genetic effects.

## 4. Conclusion

We used European and Alaskan three-spined stickleback and *S. solidus* in experimental infections and found that infection phenotypes were determined by main effects of the host and the parasite. We identified different defence mechanisms in

this system: qualitative resistance (the inverse of parasite infection success), quantitative resistance (parasite growth suppression) and tolerance (the relationship between infection intensity and measures of host health). Although qualitative resistance depended, over the scale of continents, on host–parasite interaction effects, quantitative resistance and tolerance did not. We conclude that host, parasite and interaction effects differentially affect distinct defence mechanisms.

## 5. Material and methods

### (a) Hosts and parasites

Hosts and parasites came from two European and two Alaskan populations (table 1; figure 1). European hosts and parasites are characterized by low resistance against *S. solidus* (DE stickleback) and high growth in sticklebacks (NO *S. solidus*) [31,45,46,56]. Alaskan host–parasite pairs show highly diverse infection phenotypes on a small geographical scale [38,39,57].

We used laboratory-bred first-generation offspring from wild-caught stickleback and *S. solidus*. Stickleback eggs were fertilized *in vitro* in 3 ppt artificial seawater. Alaskan stickleback eggs were rinsed with acriflavine (50 µl l$^{-1}$; 30 s) and methylene blue (500 µg l$^{-1}$ methylene blue; 30 s) and shipped to the Max Planck Institute (MPI) for Evolutionary Biology, Plön, Germany, at 4°C. German stickleback eggs were treated in the same way. A fin clip of each parent was retained for

**Table 1.** Host and parasite origins.

| ID | | sampling site | | |
|---|---|---|---|---|
| Walby | Alaskan | Walby Lake | Alaska | 61°37′N, −149°12′W |
| Wolf | Alaskan | Wolf Lake | Alaska | 61°38′N, −149°16′W |
| DE | European stickleback | Großer Plöner See | Germany | 54°08′N, 10°24′E |
| NO | European *S. solidus* | Lake Skogseidvatnet | Norway | 60°13′N, 05°53′E |

**Table 2.** Exposed and infected stickleback. Numbers denote *S. solidus*- and sham-exposed stickleback nine weeks post exposure; the number of infected stickleback is indicated in brackets.

| | Walby stickleback | | Wolf stickleback | | DE stickleback | |
|---|---|---|---|---|---|---|
| Walby *S. solidus* | 45 | (14) | 60 | (7) | 59 | (9) |
| Wolf *S. solidus* | 48 | (9) | 59 | (15) | 59 | (4) |
| NO *S. solidus* | 46 | (8) | 57 | (0) | 57 | (15) |
| Sham-exposed | 33 | (1) | 34 | | 35 | |

downstream genetic analyses. The fish were kept at the institute's aquaria system at 18°C and a light:dark cycle of 16: 8 h. They were eight months old at the start of the infection experiment. *Schistocephalus solidus* plerocercoids came from infected Alaskan fish that were shipped to the MPI and dissected immediately upon arrival. Pairs of Alaskan and European *S. solidus* plerocercoids were weight-matched and bred *in vitro* [36,58]. The eggs were kept at 4°C in the dark and incubated at 18°C for three weeks before the hatch was stimulated by light exposure [59].

## (b) Infection experiment
Laboratory-cultured copepods (*Macrocyclops albidus*) were used as first intermediate hosts and microscopically screened for procercoids one week after exposure to single coracidia. Individually housed stickleback were starved for 1 day and exposed to single-infected *M. albidus* on day 16. The fish were transferred to 16 l aquaria 2 days later. Water from each treatment group was sieved and screened for leftover copepods in order to determine the exact number of exposed fish.

The experiment was composed of three rounds. In each round, hosts from the three populations were exposed to each of the three parasite strains (i.e. *S. solidus* from a distinct location) or sham-exposed (figure 1). Parasite sibships (offspring from one pair of worms; *n* = 4 per *S. solidus* strain; figure 1*b*) were the same in every round; fish families (offspring from one pair of fish) differed between rounds. We used a common garden approach to minimize confounding factors. One tank housed fish from all populations; controls had their own compartment (figure 1*b*). Each tank (*n* = 36) housed 16 individuals in round 1 and 17 individuals in rounds 2 and 3. The fish were fed frozen chironomids three times a week. The number of fish per tank was kept constant by replacing dead individuals with naive fish from the same family. Six controls and one exposed fish died before the end of the experiment; one control fish was infected and excluded from further analyses (table 2). Stickleback were euthanized with MS222 and dissected nine weeks post exposure. At the end of the experiment 82 fish were infected, 409 fish were exposed but uninfected and 102 fish were sham-exposed (table 2).

## (c) Phenotypic measurements and tissue sampling
We recorded fish sex, standard length (±1 mm) and total weight (±0.1 mg). Head kidneys, liver and spleen were weighted to the nearest 0.1 mg. Head kidneys were immediately transferred to RNAlater (Sigma-Aldrich) and stored at −20°C. Plerocercoids were removed from the body cavity, weighted, transferred to liquid nitrogen and stored at −80°C. DNA was extracted from fin clips using the DNeasy 96 kit (Qiagen) following the manufacture's protocol. Each fish was assigned to its family by using 15 different microsatellite loci in four PCR protocols [24,60,61].

We determined the infection rate as the proportion of exposed fish (corrected by the number of copepods that have not been eaten) that became infected. The relative weight of the parasite, the PI, was calculated as $100 \times$ parasite weight/fish weight [35]. Fish condition was estimated using the CF (the ratio between observed weight and expected weight at a given length = $100 \times$ fish weight/fish length$^b$ with fish population-specific exponent $b$ [43] and the HSI, which is a measure for medium-term energy reserves [44]. The immunological activation was estimated by the SSI and the HKI [62]. HSI, SSI and HKI were calculated as $100 \times$ organ weight/fish weight.

## (d) RNA extraction and reverse transcription
Head kidney RNA was extracted with a NucleoSpin 96 kit (Macherey-Nagel) following the manufacturer's protocol, including 1% β-mercaptoethanol for tissue lysis (2 × 3 min at 30 Hz; Tissue Lyser II; Qiagen) and on-column DNA digestion. RNA concentration and purity were determined spectrophotometrically (NanoDrop1000; Thermo Scientific). All $A_{260}/A_{280}$ ratios were at least 1.98 and RNA concentrations were adjusted at 500 ng for reverse transcription. We used the Omniscript RT kit (Qiagen) according to the manual but used 0.2 μl of a 4-unit RNase inhibitor (Qiagen) per reaction. The cDNA was stored at −80°C.

## (e) Quantitative real-time PCR (RT-qPCR)
Transcription levels of 32 genes were tested using 96.96 Dynamic Array integrated fluidic circuits (IFCs) on a Biomark HD system (Fluidigm) with EvaGreen as DNA intercalating dye. cDNA was pre-amplified using TaqMan PreAmp Master Mix (Applied

Biosystems) according to the manufacturer's protocol (14 cycles). The product was diluted 1 : 5 in low TE buffer (10 mM Tris, pH 8.0, 0.1 mM EDTA). Samples of all treatments were spread across three IFCs. All targets for a given sample were included in the same run and measured in technical triplicates. Inter-run calibrators and negative controls were included on each IFC.

Targets of interest covered four putative reference genes (*b2 m*, *ubc*, *rpl13a*, *ef1a* [63]), four regulatory genes (*abtb1*, *ascl1b*, *kat2a*, *mapk13*) and 24 immune genes involved in innate immunity (*marco*, *mst1ra*, *mif*, *il-1β*, *tnfr1*, *saal1*, *tlr2*, *csf3r*, *p22^phox^*, *nkef-b*, *sla1*, *cd97*), adaptive immunity (*stat4*, *stat6*, *igm*, *cd83*, *foxp3*, *tgf-β*, *il-16*, *mhcII*, *tcr-β*) and the complement system (*cfb*, *c7*, *c9*) [32,33,46,64] (more information in electronic supplementary material, SI.5).

Melting curves were analysed with the *Fluidigm Analysis software v. 4.5.1*. Three targets (*il-1β*, *tgf-β* and *ascl1b*) were excluded from further analyses due to ambiguous melting curves. Raw data were imported into *qbase+ 3.0* (Biogazelle) [65] to assess data quality and calculate calibrated normalized relative quantities (CNRQ) [66]. The negative cut-off for technical sensitivity limit was set at cycle 28 and a 0.5 cycle variation was accepted for maximum triplicate variability. Reference targets *rpl13* and *ubc* were used for normalization as inferred from geNorm ($M = 0.236$) and the coefficient of variation ($CV = 0.082$) [65,67]. Target-specific amplification efficiencies (1.85–2.24) were calculated from a serial dilution. The data were $\log_{10}$ transformed. Missing values (one for *c9*, *cd83* and *marco*; two for *cfb* and *saal1*; five for *tcr-β*) were replaced by the mean.

## (f) Statistical analyses

All statistical analyses were performed in R (v. 3.2.0; [68]). Infection rates were analysed with binomial GLMMs using glmer() from *lme4* [69]. Response variables were proportional data from infected versus uninfected individuals. Infection rates in copepods were analysed with parasite population and experimental round and their interaction as fixed effects and parasite sibship as random intercept. Infection rates in fish were analysed with host and parasite population and their interaction as fixed effects and experimental round and parasite sibship as crossed random effects. Sympatry was used as predictor for infection rates in Alaskan fish. We accounted for the number of copepods that were not ingested. Significantly different groups and *p*-values were determined with glht() from *multcomp* [70] with individually defined contrasts or Type III Wald $\chi^2$-tests using Anova() from *car* [71].

Further analyses distinguished between (i) sham-exposed controls, (ii) *S. solidus* exposed but uninfected fish (exposed) and (iii) *S. solidus* infected fish. Linear mixed-effects models (using lmer() from *lme4*) were used to test for differences between parasite growth (PI), fish condition (CF, HSI) and immunological parameters (SSI, HKI). To avoid rank deficient fixed-effect model matrices we separated the data according to host and parasite origins because NO parasites did not infect Wolf fish. Models using data from infected fish included host or parasite origin as fixed effect as well as fish sex and tank, which is confounded with fish family and parasite sibship, as crossed random effects. We excluded data from two fish due to missing information on fish sex. Multiple testing was accounted for by using the false discovery rate (FDR) according to Benjamini & Hochberg [72] with $\alpha = 0.05$. Different slopes of the relationship between infection intensity and measures of host health or fitness indicate variance in tolerance [8,10,11]. We studied the relationship between infection intensity and body condition because body condition predicts mate quality and fitness in three-spined stickleback [73]. We fitted parasite-strain-specific linear mixed effect models (lmer() from *lmerTest* [74]) with CF or HSI as dependent variable, host population and PI and their interaction as fixed effects and fish sex and tank as random effects. The

corresponding degrees of freedom were approximated with Satterthwaite's method.

Gene expression profiles were analysed with a multivariate approach by grouping data from all 25 targets (*total*), 11 innate immune genes (*innate*: *marco*, *mst1ra*, *mif*, *tnfr1*, *saal1*, *tlr2*, *csf3r*, *p22^phox^*, *nkef-b*, *sla1*, *cd97*), eight adaptive immune genes (*adaptive*: *stat4*, *stat6*, *igm*, *cd83*, *foxp3*, *il-16*, *mhcII*, *tcr-β*), three complement component genes (*complement*: *c7*, *c9*, *cfb*) or three regulatory genes (*regulatory*: *abtb1*, *kat2a*, *mapk13*). Non-parametric permutational multivariate analyses of variance (PERMANOVA [75]) were based on Euclidian distances and 10 000 permutations that were constrained within tank. The weight of the fish was included as covariate to account for size related effects. Pairwise PERMANOVAs were used *a posteriori* to identify significantly different groups [75]. We tested for differences in baseline gene expression by using data from sham-exposed controls of the three populations. We tested whether host, parasite and/or their interaction affected gene expression of *S. solidus* exposed stickleback. Since NO parasites did not infect hosts from Wolf, we grouped data from (i) each parasite population and (ii) from each host population (electronic supplementary material, SI.4.2).

Gene expression profiles of infected, exposed and control fish were compared within each combination of hosts and parasites. We used the FDR to account for multiple testing [72]. Differentially expressed single genes were identified by use of linear mixed effect models with tank as random intercept. Plots were created with *ggplot2* [76] and *aheatmap*() from *NMF* [77].

**Ethics.** Animal experiments were approved by the Ministry of Energy Transition, Agriculture, the Environment and Rural Areas of the state Schleswig-Holstein, Germany (reference number: V 312'7224.123-34). Use of live Alaskan stickleback was approved by the Stony Brook University Institutional Animal Care and Use Committee (IACUC reference number: 237429-18).

**Data accessibility.** All data and code are publicly accessible on EDMOND, the Open Access Data Repository of the Max Planck Society, at https://doi.org/10.17617/3.4u [78]. Supplementary information is provided in the electronic supplementary material [79].

**Authors' contributions.** A.P.: conceptualization, data curation, formal analysis, investigation, methodology, visualization, writing—original draft, writing—review and editing; M.A.H.: data curation, formal analysis, investigation, methodology, writing—review and editing; O.R.: resources, software, supervision, validation, writing—review and editing; N.M.D.: conceptualization, data curation, formal analysis, funding acquisition, project administration, supervision, writing—review and editing; D.C.H.: conceptualization, formal analysis, funding acquisition, project administration, resources, supervision, validation, writing—review and editing; M.A.B.: conceptualization, funding acquisition, methodology, project administration, resources, supervision, writing—review and editing; M.K.: conceptualization, funding acquisition, investigation, methodology, project administration, resources, supervision

All authors gave final approval for publication and agreed to be held accountable for the work performed therein.

**Competing interests.** We declare we have no competing interests.

**Funding.** Open access funding provided by the Max Planck Society.

A.P. was financially supported by the International Max Planck Research School for Evolutionary Biology. O.R. was funded by ERC Starting Grant MALEPREG (grant no. 755659) and grants from the German Research Foundation (grant nos. 349393951, 237263721 and 274695381). N.M.D. received grants from Eppley Foundation for Research and the Laurie Landeau Foundation LLC. D.C.H. was supported by the Newcomb Institute. M.A.B. was supported by a grant from the US National Institute of General Medical Sciences grant no. R01GM124330. M.K. was funded by the German Science Foundation (Priority programme 1399).

**Acknowledgements.** Martin Kalbe was central to development of this project, but sadly he died before submission of this paper. Martin was a dear friend and valued colleague. We mourn his untimely loss and dedicate this paper to his memory. We are grateful to Withe Derner, Daniel Martens, Ralf Schmuck, Gisela Schmiedeskamp, Ines Schulz, Michael Schwarz and Nina Wildenhayn for practical

support and animal husbandry. We thank Tina Henrich for helpful discussion and support. Primer sequences for *abtb1, ascl1b, kat2a, mapk13* were kindly provided by Jakob Gismann and Melanie Heckwolf.

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
