## [Peer Review File · Proceedings of the Royal Society B: Biological Sciences]

Review History

RSPB-2020-2495.R0 (Original submission)

Review form: Reviewer 1

Recommendation

Major revision is needed (please make suggestions in comments)

Scientific importance: Is the manuscript an original and important contribution to its field?

Good

General interest: Is the paper of sufficient general interest?

Good

Quality of the paper: Is the overall quality of the paper suitable?

Good

Is the length of the paper justified?

Yes

Should the paper be seen by a specialist statistical reviewer?

No

Do you have any concerns about statistical analyses in this paper? If so, please specify them explicitly in your report.

No

It is a condition of publication that authors make their supporting data, code and materials available - either as supplementary material or hosted in an external repository. Please rate, if applicable, the supporting data on the following criteria.

Is it accessible?

Yes

Is it clear?

Yes

Is it adequate?

Yes

Do you have any ethical concerns with this paper?

No

Comments to the Author

This manuscript focuses on understanding the hosts, parasites, and their interactive effects on epidemiological traits. They use stickleback and *Schistocephalus solidus* as their host-parasite system. They utilize three different host and stickleback populations and conduct experimental cross infections to look at qualitative resistance, quantitative resistance, and tolerance. They also investigate how infection alters gene expression profiles. The authors find that qualitative resistance (infection rate) was determined by the interaction between hosts and parasite genotypes, whereas quantitative resistance and tolerance depended primarily on host genotype. Additionally, while there were genotype specific gene expression profiles of uninfected stickleback, gene expression converged upon infection.

This study, for the most part, is well designed, straightforward, and detailed. However, there are certain methodological issues that need to be addressed to determine if the results are robust.

Major comments:

Do you have any idea how one of your sham-exposed control got infected? I think there needs to be more of a justification of what might have happened and why you do not believe any other fish were affected. My initial reaction is that if the sham-exposed control is infected, then the whole tank may be compromised and that simply excluding that one control fish (P20 L422) may not be sufficient to fix the problem.

In general, the authors should go into more detail about measuring tolerance and what results would signify a tolerant population. For instance, you should explain more about the tolerance results in discussion (P16 304-305) and what slopes/changes in slope would indicate more tolerant populations in the methods (P23 L 511-513).

Why did you not use either DE stickleback and parasites or NO stickleback and parasites? There needs to be a justification of why you used DE stickleback and NO parasites, especially since these sites are very far apart.

P19 L406-408: Stickleback were individually housed and given an infected copepod. If the stickleback did not eat the copepod (which you determined by sieving the leftover copepods), did you include those stickleback in the experiment? I do not think they should be included in the study. Were these stickleback included in the study? If so, how did you account for them, since

they could not have gotten infected? You mention on P23 L495-6 that “ we accounted for the number of copepods that were not ingested.” I don’t think this is the correct approach, and believe that these individuals should not have been included in analyses. If they are, there should be clear justification, which I do not see in the text.

The authors should spend more time in the discussion putting their results in a larger context. How does this work compare with previous work in stickleback? What is novel about this study?

Minor Comments:

P7 L140-141. Why isn’t host population included as a significant factor ($p = 0.0206$)? I’m guessing it isn’t significant after post *fdr* correction. However, this is not indicated in the legend for Table S1.

P7 L146-147: I found this sentence confusing. Perhaps this instead: “Parasite indices (PI), approximations of host quantitative resistance, differed between host populations, but were not affected by parasite strains.

P7 L148: You should mention that DE-Walby contrast for wolf parasites was not significant after *fdr* correction.

P9 L177-181: Where are the analysis outputs for these stats? Please add supplementary table number if they are there.

Supplementary Figures – Many of the supplementary figures (S2, S3, etc) are using wrong/old labels for these populations. Please change for consistency.

Figure S3 – I find it very hard to compare control and treatment values in this figure. Perhaps you could move the controls down with the rest of the data and indicate that the controls are the same for given host population.

P13 L263 – Name the four adaptive genes.

P17 L 361-364 – Reword. “*S. Solidus* can control stickleback gene expression” is misleading since many parasites can actively control host behavior/phenotype for their own fitness. In this case, I think it would be more apt to say that certain *S. Solidus* strains cause convergent gene expression across different stickleback populations (or something to that effect).

P18 L383 – remove “host” from “hosthost”

P18 L385 – Add citation(s)

Define parasite sibships and fish families.

P19 L417 – Figure 1 shows 17 individuals per tank. For round 1, which group had one less fish (leading to 16 fish per tank)?

Table 2 – This table shows the total number of infected stickleback not the proportion.

P21 L463 – Write out what an IFC is.

Review form: Reviewer 2

Recommendation

Accept with minor revision (please list in comments)

Scientific importance: Is the manuscript an original and important contribution to its field?

Excellent

General interest: Is the paper of sufficient general interest?

Excellent

Quality of the paper: Is the overall quality of the paper suitable?

Excellent

Is the length of the paper justified?

Yes

Should the paper be seen by a specialist statistical reviewer?

No

Do you have any concerns about statistical analyses in this paper? If so, please specify them explicitly in your report.

No

It is a condition of publication that authors make their supporting data, code and materials available - either as supplementary material or hosted in an external repository. Please rate, if applicable, the supporting data on the following criteria.

Is it accessible?

Yes

Is it clear?

Yes

Is it adequate?

Yes

Do you have any ethical concerns with this paper?

No

Comments to the Author

Using experimental cross-infections of three-spined sticklebacks and their *S. solidus* parasites from Alaskan and European populations, the authors address host defence mechanisms in terms of qualitative and quantitative resistance and tolerance to explore genotypic and phenotypically plastic effects on epidemiological traits of the host-parasite interaction. The study has a well-taken and strong approach with a good experimental setup. The paper is well written and easy to follow and the conclusions appear sound and justifiable. I only have some minor comments and suggestions:

- l. 368-369: This statement is unclear to me. Please rephrase for clarity!

- l. 383: Change 'hosthost' to 'host'

- l. 393: Include 'Max Planck Institute' to explain the acronym 'MPI'

- l. 428: Replace 'proportion' with 'number'

- l. 442-446: Explain PI, CF, HIS, SSI and HKI in detail and provide the equations required to calculate these indices! The reader should not have to look this up in the various references.

- l. 490: Replace 'round' with 'experimental round'

- 1. 495-496: "We accounted for the number of copepods that were not ingested" - How was this done?

Review form: Reviewer 3

Recommendation

Accept with minor revision (please list in comments)

Scientific importance: Is the manuscript an original and important contribution to its field?

Acceptable

General interest: Is the paper of sufficient general interest?

Acceptable

Quality of the paper: Is the overall quality of the paper suitable?

Excellent

Is the length of the paper justified?

Yes

Should the paper be seen by a specialist statistical reviewer?

No

Do you have any concerns about statistical analyses in this paper? If so, please specify them explicitly in your report.

No

It is a condition of publication that authors make their supporting data, code and materials available - either as supplementary material or hosted in an external repository. Please rate, if applicable, the supporting data on the following criteria.

Is it accessible?

Yes

Is it clear?

Yes

Is it adequate?

Yes

Do you have any ethical concerns with this paper?

No

Comments to the Author

This is a very well-presented study that aims to determine how qualitative resistance, quantitative resistance and tolerance of sticklebacks to tapeworm parasites varies in host/parasite combinations from Alaska and Europe. I found the experiment to be well-conducted, the analyses appropriate and the dense array of results well-presented. I only really have one major point that I would like to see addressed, in relation to whether the results are a broad property of host-parasite systems, or whether they're a property of the populations that happen to have been sampled here. Otherwise, I think this is a nice study.

How confident can you be that you would have got the same results if you had used different populations of fish and worms? In other words, are the observed results likely to be a general property of hosts/parasites from different locations, or is it just because you happened to choose these populations? For example, according to lines 381-383, the DE fish have low resistance and the NO worms are highly virulent...would you have got the same results with a highly resistant European fish and a less virulent European worm?

Lines 381-383: presumably when you say that DE fish have low resistance, you mean to a specific population of worms (presumably a Danish population)? Similarly when saying that NO worms have high growth...or has this been shown to be the case to multiple populations of worms and in multiple populations of fish....? I guess what I'm saying is that you're contending that you've picked a really susceptible European fish and a really virulent European worm, and yet the whole aim of the study is to find out whether this is always the case or whether it depends on what the combination of host and worm is.

Table 2: the numbers in brackets are supposedly proportions, but proportions should be between 0 and 1: are these percentages? This needs clarifying.

Lines 467-472: how were these genes chosen? I think it important to state this briefly. Off the top of my head and I can see the functional relevance of some of them, but not others, so it might be useful to add a supplementary table stating the function of each gene or the pathways that it is associated with – e.g. you could say that *foxp3* is associated with a regulatory phenotype.

Decision letter (RSPB-2020-2495.R0)

18-Jan-2021

Dear Dr Piecyk:

I am writing to inform you that your manuscript RSPB-2020-2495 entitled "Experimental infections reveal distinct defence mechanisms in European and Alaskan populations of the three-spined stickleback *Gasterosteus aculeatus*" has, in its current form, been rejected for publication in Proceedings B.

This action has been taken on the advice of referees, who have recommended that substantial revisions are necessary. With this in mind we would be happy to consider a resubmission, provided the comments of the referees are fully addressed. However please note that this is not a provisional acceptance.

- 1) A 'response to referees' document including details of how you have responded to the comments, and the adjustments you have made.
- 2) A clean copy of the manuscript and one with 'tracked changes' indicating your 'response to referees' comments document.

- 3) Line numbers in your main document.
 4) Data - please see our policies on data sharing to ensure that you are complying (<https://royalsociety.org/journals/authors/author-guidelines/#data>).

Sincerely,
 Professor Hans Heesterbeek
 mailto: proceedingsb@royalsociety.org

Associate Editor
 Board Member: 1
 Comments to Author:

Thank you for giving Proc B the opportunity to review this interesting, carefully conducted, and well-presented study. The reviewers agree that the MS has a lot of potential, but there are several major concerns that would need to be addressed in order for Proc B to consider a resubmission, outlined by the reviewers below. Please pay special attention to the following: (1) Why was one of the sham-exposed fish found to be infected and how can you be sure that the rest of the fish in that tank were not compromised? (2) Did you exclude stickleback that were given an infected copepod but did not eat it? If not, either exclude those individuals and re-run analyses or justify your choice. (3) Please provide an expanded commentary on the potential generality of your results.

Reviewer(s)' Comments to Author:
 Referee: 1
 Comments to the Author(s)

This manuscript focuses on understanding the hosts, parasites, and their interactive effects on epidemiological traits. They use stickleback and *Schistocephalus solidus* as their host-parasite system. They utilize three different host and stickleback populations and conduct experimental cross infections to look at qualitative resistance, quantitative resistance, and tolerance. They also investigate how infection alters gene expression profiles. The authors find that qualitative resistance (infection rate) was determined by the interaction between hosts and parasite genotypes, whereas quantitative resistance and tolerance depended primarily on host genotype. Additionally, while there were genotype specific gene expression profiles of uninfected stickleback, gene expression converged upon infection.

This study, for the most part, is well designed, straightforward, and detailed. However, there are certain methodological issues that need to be addressed to determine if the results are robust.

Major comments:

Do you have any idea how one of your sham-exposed control got infected? I think there needs to be more of a justification of what might have happened and why you do not believe any other fish were affected. My initial reaction is that if the sham-exposed control is infected, then the whole tank may be compromised and that simply excluding that one control fish (P20 L422) may not be sufficient to fix the problem.

In general, the authors should go into more detail about measuring tolerance and what results would signify a tolerant population. For instance, you should explain more about the tolerance results in discussion (P16 304-305) and what slopes/changes in slope would indicate more tolerant populations in the methods (P23 L 511-513).

Why did you not use either DE stickleback and parasites or NO stickleback and parasites? There needs to be a justification of why you used DE stickleback and NO parasites, especially since these sites are very far apart.

P19 L406-408: Stickleback were individually housed and given an infected copepod. If the stickleback did not eat the copepod (which you determined by sieving the leftover copepods), did you include those stickleback in the experiment? I do not think they should be included in the study. Were these stickleback included in the study? If so, how did you account for them, since they could not have gotten infected? You mention on P23 L495-6 that “ we accounted for the number of copepods that were not ingested.” I don’t think this is the correct approach, and believe that these individuals should not have been included in analyses. If they are, there should be clear justification, which I do not see in the text.

The authors should spend more time in the discussion putting their results in a larger context. How does this work compare with previous work in stickleback? What is novel about this study?

Minor Comments:

P7 L140-141. Why isn’t host population included as a significant factor ($p = 0.0206$)? I’m guessing it isn’t significant after post *fdr* correction. However, this is not indicated in the legend for Table S1.

P7 L146-147: I found this sentence confusing. Perhaps this instead: “Parasite indices (PI), approximations of host quantitative resistance, differed between host populations, but were not affected by parasite strains.

P7 L148: You should mention that DE-Walby contrast for wolf parasites was not significant after *fdr* correction.

P9 L177-181: Where are the analysis outputs for these stats? Please add supplementary table number if they are there.

Supplementary Figures – Many of the supplementary figures (S2, S3, etc) are using wrong/old labels for these populations. Please change for consistency.

Figure S3 – I find it very hard to compare control and treatment values in this figure. Perhaps you could move the controls down with the rest of the data and indicate that the controls are the same for given host population.

P13 L263 – Name the four adaptive genes.

P17 L 361-364 – Reword. “S Solidus can control stickleback gene expression” is misleading since many parasites can actively control host behavior/phenotype for their own fitness. In this case, I think it would be more apt to say that certain S. Solidus strains cause convergent gene expression across different stickleback populations (or something to that effect).

P18 L383 – remove “host” from “hosthost”

P18 L385 – Add citation(s)

Define parasite sibships and fish families.

P19 L417 – Figure 1 shows 17 individuals per tank. For round 1, which group had one less fish (leading to 16 fish per tank)?

Table 2 – This table shows the total number of infected stickleback not the proportion.

P21 L463 – Write out what an IFC is.

Referee: 2

Comments to the Author(s)

Using experimental cross-infections of three-spined sticklebacks and their *S. solidus* parasites from Alaskan and European populations, the authors address host defence mechanisms in terms of qualitative and quantitative resistance and tolerance to explore genotypic and phenotypically plastic effects on epidemiological traits of the host-parasite interaction. The study has a well-taken and strong approach with a good experimental setup. The paper is well written and easy to follow and the conclusions appear sound and justifiable. I only have some minor comments and suggestions:

- l. 368-369: This statement is unclear to me. Please rephrase for clarity!
- l. 383: Change 'hosthost' to 'host'
- l. 393: Include 'Max Planck Institute' to explain the acronym 'MPI'
- l. 428: Replace 'proportion' with 'number'
- l. 442-446: Explain PI, CF, HIS, SSI and HKI in detail and provide the equations required to calculate these indices! The reader should not have to look this up in the various references.
- l. 490: Replace 'round' with 'experimental round'
- l. 495-496: "We accounted for the number of copepods that were not ingested" – How was this done?

Referee: 3

Comments to the Author(s)

This is a very well-presented study that aims to determine how qualitative resistance, quantitative resistance and tolerance of sticklebacks to tapeworm parasites varies in host/parasite combinations from Alaska and Europe. I found the experiment to be well-conducted, the analyses appropriate and the dense array of results well-presented. I only really have one major point that I would like to see addressed, in relation to whether the results are a broad property of host-parasite systems, or whether they're a property of the populations that happen to have been sampled here. Otherwise, I think this is a nice study.

How confident can you be that you would have got the same results if you had used different populations of fish and worms? In other words, are the observed results likely to be a general property of hosts/parasites from different locations, or is it just because you happened to choose these populations? For example, according to lines 381-383, the DE fish have low resistance and the NO worms are highly virulent...would you have got the same results with a highly resistant European fish and a less virulent European worm?

Lines 381-383: presumably when you say that DE fish have low resistance, you mean to a specific population of worms (presumably a Danish population)? Similarly when saying that NO worms have high growth...or has this been shown to be the case to multiple populations of worms and in multiple populations of fish....? I guess what I'm saying is that you're contending that you've picked a really susceptible European fish and a really virulent European worm, and yet the whole aim of the study is to find out whether this is always the case or whether it depends on what the combination of host and worm is.

Table 2: the numbers in brackets are supposedly proportions, but proportions should be between 0 and 1: are these percentages? This needs clarifying.

Lines 467-472: how were these genes chosen? I think it important to state this briefly. Off the top of my head and I can see the functional relevance of some of them, but not others, so it might be

useful to add a supplementary table stating the function of each gene or the pathways that it is associated with – e.g. you could say that *foxp3* is associated with a regulatory phenotype.

Author's Response to Decision Letter for (RSPB-2020-2495.R0)

See Appendix A.

RSPB-2021-1758.R0

Review form: Reviewer 2

Recommendation

Accept as is

Scientific importance: Is the manuscript an original and important contribution to its field?

Excellent

General interest: Is the paper of sufficient general interest?

Excellent

Quality of the paper: Is the overall quality of the paper suitable?

Excellent

Is the length of the paper justified?

Yes

Should the paper be seen by a specialist statistical reviewer?

No

Do you have any concerns about statistical analyses in this paper? If so, please specify them explicitly in your report.

No

It is a condition of publication that authors make their supporting data, code and materials available - either as supplementary material or hosted in an external repository. Please rate, if applicable, the supporting data on the following criteria.

Is it accessible?

Yes

Is it clear?

Yes

Is it adequate?

Yes

Do you have any ethical concerns with this paper?

No

Comments to the Author

I'm satisfied with the revised version of the paper and have no further comments.

Review form: Reviewer 3

Recommendation

Accept as is

Scientific importance: Is the manuscript an original and important contribution to its field?

Good

General interest: Is the paper of sufficient general interest?

Acceptable

Quality of the paper: Is the overall quality of the paper suitable?

Good

Is the length of the paper justified?

Yes

Should the paper be seen by a specialist statistical reviewer?

No

Do you have any concerns about statistical analyses in this paper? If so, please specify them explicitly in your report.

No

It is a condition of publication that authors make their supporting data, code and materials available - either as supplementary material or hosted in an external repository. Please rate, if applicable, the supporting data on the following criteria.

Is it accessible?

Yes

Is it clear?

Yes

Is it adequate?

Yes

Do you have any ethical concerns with this paper?

No

Comments to the Author

The revised manuscript is a strong piece of work in my opinion.

Decision letter (RSPB-2021-1758.R0)

31-Aug-2021

Dear Dr Piecyk

I am pleased to inform you that your Review manuscript RSPB-2021-1758 entitled "Cross-continental experimental infections reveal distinct defence mechanisms in populations of the three-spined stickleback *Gasterosteus aculeatus*" has been accepted for publication in Proceedings B.

The referees do not recommend any further changes. Therefore, please proof-read your manuscript carefully and upload your final files for publication. Because the schedule for publication is very tight, it is a condition of publication that you submit the revised version of your manuscript within 7 days. If you do not think you will be able to meet this date please let me know immediately.

To upload your manuscript, log into <http://mc.manuscriptcentral.com/prsb> and enter your Author Centre, where you will find your manuscript title listed under "Manuscripts with Decisions." Under "Actions," click on "Create a Revision." Your manuscript number has been appended to denote a revision.

You will be unable to make your revisions on the originally submitted version of the manuscript. Instead, upload a new version through your Author Centre.

1) A text file of the manuscript (doc, txt, rtf or tex), including the references, tables (including captions) and figure captions. Please remove any tracked changes from the text before submission. PDF files are not an accepted format for the "Main Document".

2) A separate electronic file of each figure (tiff, EPS or print-quality PDF preferred). The format should be produced directly from original creation package, or original software format. Please note that PowerPoint files are not accepted.

3) Electronic supplementary material: this should be contained in a separate file from the main text and the file name should contain the author's name and journal name, e.g. `authurname_procb_ESM_figures.pdf`

All supplementary materials accompanying an accepted article will be treated as in their final form. They will be published alongside the paper on the journal website and posted on the online figshare repository. Files on figshare will be made available approximately one week before the accompanying article so that the supplementary material can be attributed a unique DOI. Please see: <https://royalsociety.org/journals/authors/author-guidelines/>

4) Data-Sharing and data citation

It is a condition of publication that data supporting your paper are made available. Data should be made available either in the electronic supplementary material or through an appropriate repository. Details of how to access data should be included in your paper. Please see <https://royalsociety.org/journals/ethics-policies/data-sharing-mining/> for more details.

<http://datadryad.org/submit?journalID=RSPB&manu=RSPB-2021-1758> which will take you to your unique entry in the Dryad repository.

Once again, thank you for submitting your manuscript to Proceedings B and I look forward to receiving your final version. If you have any questions at all, please do not hesitate to get in touch.

Sincerely,
Professor Hans Heesterbeek
mailto:proceedingsb@royalsociety.org

Associate Editor

Comments to Author:

The authors have done a fantastic job of addressing the prior round of comments. I'm very pleased to accept this MS for publication in Proc B.

Reviewer(s)' Comments to Author:

Referee: 3

Comments to the Author(s).

The revised manuscript is a strong piece of work in my opinion.

Referee: 2

Comments to the Author(s).

I'm satisfied with the revised version of the paper and have no further comments.

Decision letter (RSPB-2021-1758.R1)

02-Sep-2021

Dear Dr Piecyk

I am pleased to inform you that your manuscript entitled "Cross-continental experimental infections reveal distinct defence mechanisms in populations of the three-spined stickleback *Gasterosteus aculeatus*" has been accepted for publication in Proceedings B.

Data Accessibility section

Open Access

Paper charges

Sincerely,

Proceedings B

Appendix A

Max Planck Institute for Evolutionary Biology

MAX-PLANCK-GESELLSCHAFT

Dr. Agnes Piecyk
MPI | Max Planck Institute for Evolutionary
Biology
Dept Evolutionary Ecology; Parasitology Group
August-Thienemann Str. 2, 24306 Plön
Germany

GEOMAR | Helmholtz Centre for Ocean
Research Kiel Düsternbrookerweg 20, 24105
Kiel, Germany

apiecyk@evolbio.mpg.de

5. August 2021

Dear Professor Heesterbeek,
dear Editorial Office,
dear Referees,

This is a resubmission under the reference number RSPB-2020-2495.

We thank you for considering our manuscript “Experimental infections reveal distinct defence mechanisms in European and Alaskan populations of the three-spined stickleback *Gasterosteus aculeatus*” for publication in *Proceeding of the Royal Society B*.

We are grateful for the careful evaluation and the positive feedback and herewith submit a revised version in which we fully addressed and incorporated all Editorial and Referee comments. We have strengthened and clarified the manuscript by addressing methodological issues and putting our results in a larger context. Our elaboration in the main manuscript is concise in order to pinpoint the most important aspects and stay within the word limitation. More detailed information is stated in the point-by-point responses below.

The line numbers refer to line numbers of the clean copy of the manuscript (without Markups). Thank you again for giving us the chance of a resubmission. We are confident that our changes incorporate all comments comprehensively and are very much looking forward to hearing back from you

With kind regards and on behalf of all authors,

Agnes Piecyk

Associate Editor (Board Member: 1): comments and responses

Thank you for giving Proc B the opportunity to review this interesting, carefully conducted, and well-presented study. The reviewers agree that the MS has a lot of potential, but there are several major concerns that would need to be addressed in order for Proc B to consider a resubmission, outlined by the reviewers below. Please pay special attention to the following:

(1) Why was one of the sham-exposed fish found to be infected and how can you be sure that the rest of the fish in that tank were not compromised?

Response: *In order to give you a better visualization of our approach, Figure 1 on this page shows pictures of the infection experiment. We used 2 L aquaria to expose single fish to single copepods: each stickleback was isolated in a 2 L aquarium and starved for one day. We fed the fish with either one uninfected (control) copepod or one singly infected copepod and then covered the aquaria (left panel and upper right). The labels on the aquaria denote (a) the fish population and (b) a number indicating the parasite sibship. One control fish might have been mistakenly exposed to an infected copepod at this stage. However, since this approach has been well established in our laboratory for many years, this must have been an exception. Two days later, the fish were transferred to their respective 16 L aquaria (lower right panel), and there was no evidence that the rest of the fish in the tank were infected. Hence, we are confident the other fish were not compromised.*

Figure 1. Fish exposure. (Clockwise starting left) 2 L aquaria were used to expose single sticklebacks to single infected or uninfected copepods. The tanks were covered and fish from three different origins were transferred to 16 L aquaria.

(2) Did you exclude stickleback that were given an infected copepod but did not eat it? If not, either exclude those individuals and re-run analyses or justify your choice.

Response: *We accounted for the number of fish that did not eat an infected copepod by sieving the water of the single tanks in which the fish were exposed. We used one filter per treatment group (i.e. fish family x worm sibship combination) because using one filter per individual would have been too time consuming causing either smaller sample sizes or more experimental rounds. Increasing the number of experimental rounds would have caused sample sizes that would have been too small for a thorough investigation of round effects. Thus, we did not know which individual did not eat the copepod and, therefore, no fish was excluded. However, we were able to account for leftover copepods in the statistical models by using the corrected number of exposed fish per treatment group (number of exposed fish minus number of leftover copepods).*

(3) Please provide an expanded commentary on the potential generality of your results.

Response: *The presented experiment was designed to study infection outcomes in cross-continental helminth infections. We aimed to investigate the effect of time (along the infection process) and space (geography). We are confident that our findings complement previous studies on sticklebacks showing that infection success can be linked to geography and, presumably, phylogeny of the host and the parasite (e.g., in Weber et al., 2016; Piecyk et al., 2019a), but that, importantly, environmental variation is also a causal factor in immunological heterogeneity. The fact that the two Alaskan hosts and parasites showed different infection success, even though the lakes are in close proximity, emphasizes the effect of the environment on co-evolutionary trajectories and infection outcomes. The importance of considering the effect of environmental variation on the evolution of immunologically heterogeneous populations and infection outcomes is stated more explicitly in the revised manuscript:*

“Whether inter-continental resistance can be attributed to local adaptation or specificities of host and parasite populations or clades, warrants further investigation that takes the effect of environmental variation on defence mechanisms and infection outcomes into consideration.” (Lines 313-317)

Population divergence, here indicated by different baseline condition and immunological activation, has previously been shown for various lake versus river fish and linked to parasite prevalence and abundance (Huang et al., 2016). Moreover, expression plasticity (here: the effect that differences of (immune) gene expression converge upon infection) seems to be a general property of sticklebacks, as it has been shown in experiments using sticklebacks from Vancouver Island, Canada (Stutz et al., 2015; Lohman et al., 2017), for sticklebacks from Scotland (Robertson et al., 2015) and for DE sticklebacks infected with tapeworms from across the Northern Hemisphere (Piecyk et al. 2019a). The novelty of this study is that the expression plasticity was studied in controlled experimental infections, revealing an important effect of the parasite.

We are confident that host-parasite genotype specific qualitative resistance and gene expression, host genotype specific quantitative resistance and tolerance, as well as converging immune gene expression upon infection are general properties of this important model system and most probably other systems as well. This study presents basic research that has important implications for applied clinical sciences. Using this model

system, we could, for the first time, show that measures of host immunity (resistance, tolerance, health) need to distinguish among immune response mechanisms and be put into a co-evolutionary context involving time and space.

Referee 1: major comments, responses and respective changes

1. Do you have any idea how one of your sham-exposed control got infected? I think there needs to be more of a justification of what might have happened and why you do not believe any other fish were affected. My initial reaction is that if the sham-exposed control is infected, then the whole tank may be compromised and that simply excluding that one control fish (P20 L422) may not be sufficient to fix the problem.

Response: *In order to give you a better visualization of our approach, Figure 1 on page 2 shows pictures of the infection experiment. We used 2 L aquaria to expose single fish to single copepods: each stickleback was isolated in a 2 L aquarium and starved for one day. We fed the fish with either one uninfected (control) copepod or one singly infected copepod and covered the aquaria (left panel and upper right). The labels on the aquaria denote (a) the fish population and (b) a number indicating the parasite sibship. One control fish might have been mistakenly exposed to an infected copepod at this stage. However, since this approach has been well established in our laboratory for many years, this must have been an exception. After two days, the fish were transferred to their respective 16 L aquaria (lower right panel), and we are confident that the rest of the fish in the tank were not compromised because there was no evidence of any additional infections.*

2. In general, the authors should go into more detail about measuring tolerance and what results would signify a tolerant population. For instance, you should explain more about the tolerance results in discussion (P16 304-305) and what slopes/changes in slope would indicate more tolerant populations in the methods (P23 L 511-513).

Response: *Conceptual information on ‘tolerance’ is summarized in the Introduction (lines 58-66). Our results on tolerance were mentioned twice in the first paragraph of the Discussion section but only addressed in detail in the paragraph below. To address this concern, we structured this section more clearly in the revised version by stating population-specific tolerance once in the summarizing paragraph of the Discussion (line 293-294) and elaborating on this finding later (lines 324-337). We now also explicitly mention the slopes in the Results (lines 177-178).*

“Resistance reduces the likelihood of infection (qualitative resistance) or limits parasite replication or growth (quantitative resistance), whereas tolerance limits the negative effects of a given parasite burden without reducing parasite replication or growth [4,8–11]. Resistance and tolerance are not mutually exclusive but have different ecological and evolutionary consequences [8,12,13]. For example, parasite prevalence is expected to decrease if hosts evolve resistance, whereas parasite prevalence may increase if hosts evolve tolerance [14,15].” (Lines 58-66)

“Although the CF decreased with increasing PI in Walby and DE fish, Wolf condition was not affected by PI (Figure 3).” (Lines 177-178)

“Our first key finding was that resistance and tolerance differed among host populations, implying host genetic effects on infection outcome.” (Lines 293-394)

“The relationship of parasite size and host condition was used to estimate tolerance. In addition to the qualitative resistance of Wolf stickleback, these fish also appeared to be more tolerant than Walby and DE hosts (Figure 3). Accordingly, stickleback populations (here, Wolf) can have both higher qualitative resistance and tolerance in comparison to sticklebacks from other, even nearby, populations. We suggest that high tolerance is a universal property of Wolf fish, whereas the prevention of infection is NO *S. solidus*-specific. Notably, *S. solidus* size depends on the size of the stickleback [47,48], and *vice versa*, and the relative contribution of environmentally mediated phenotypic plasticity to infection phenotypes can be substantial [49,50]. Accordingly, what manifests as ‘tolerance’ could result from low host condition causing low parasite growth. Furthermore, the lack of an ecological context in laboratory experiments could obscure our results.” (Lines 324-337)

3. Why did you not use either DE stickleback and parasites or NO stickleback and parasites? There needs to be a justification of why you used DE stickleback and NO parasites, especially since these sites are very far apart.

Response: *DE stickleback and NO S. solidus represent two extreme forms, thus, two reference points, of hosts and parasites in this model system: DE stickleback are usually very susceptible to S. solidus infections, whereas NO S. solidus are highly virulent (as estimated by the parasite index). This information can be found in lines 373-376 in the Material and Methods:*

“European hosts and parasites are characterised by low resistance against *S. solidus* (DE stickleback) and high growth in sticklebacks (NO *S. solidus*) [31,45,46,56].” (lines 376-378)

Moreover, S. solidus prevalence is extremely low in DE and we were not able to sample parasites from DE for this study. However, DE S. solidus have low parasite indices in sympatric fish (Scharsack et al., 2016); and S. solidus from a nearby population grow as small as those from DE in both German and Norwegian sticklebacks, linking the growth of the parasite to the parasite phylogeny (thus, parasite genotype) rather than interaction effects (see point 8 of your minor comments). Unfortunately, due to word limitations of the journal, we cannot elaborate on this in the manuscript. More information on the two European populations and their characteristics as reference points can be found in the cited literature, especially in references 31 and 46 (Piecyk et al., 2019a and 2019b) as well as in Scharsack et al. 2016 (showing the data from DE S. solidus infections).

4. P19 L406-408: Stickleback were individually housed and given an infected copepod. If the stickleback did not eat the copepod (which you determined by sieving the leftover copepods), did you include those stickleback

in the experiment? I do not think they should be included in the study. Were these stickleback included in the study? If so, how did you account for them, since they could not have gotten infected? You mention on P23 L495-6 that “ we accounted for the number of copepods that were not ingested.” I don’t think this is the correct approach, and believe that these individuals should not have been included in analyses. If they are, there should be clear justification, which I do not see in the text.

Response: *We used 2 L aquaria to expose single fish to single copepods: each stickleback was isolated in a 2 L aquarium and starved for one day. We fed the fish with either one uninfected (control) copepod or one singly infected copepod and waited for two days. In order to determine whether the fish ate the copepod, we sieved the water of the single tanks and screened the filters for leftovers. We did not use one filter per tank but pooled the leftovers from each treatment group (i.e. fish family x worm sibship combination). Thus, we could not know which individual fish did not eat the copepod. However, we accounted for leftover copepods in the statistical models on the level of the treatment group. For example, if we exposed 5 fish and found one copepod in the filter, we corrected the number of exposed fish to ‘four’. This information is stated more clearly in the revised manuscript:*

“Water from each treatment group was sieved and screened for leftover copepods in order to determine the exact number of exposed fish.” (Lines 403-405)

“We determined the infection rate as the proportion of exposed fish (corrected by the number of copepods that have not been eaten) that became infected.” (Lines 437-439)

5. The authors should spend more time in the discussion putting their results in a larger context. How does this work compare with previous work in stickleback? What is novel about this study?

Response: *Our experiment was designed to study infection outcomes in cross-continental helminth infections. We aimed to investigate the effect of time (along the infection process) and space (geography) on infection outcomes. Our findings complement previous work showing that infection success can be linked to geography and, presumably, phylogeny of the host and the parasite (e.g., in Weber et al., 2016; Piecyk et al., 2019a), but that, importantly, environmental variation also causes immunological heterogeneity. Baseline differences and the fact that the two Alaskan hosts and parasites showed different infection success, even though the lakes are in close proximity, emphasizes the effect of the environment on co-evolutionary trajectories and infection phenotypes.*

In this context, higher baseline immunological activation (as determined for DE fish) has previously been shown for various lake versus river fish and linked to parasite prevalence and abundance (Huang et al., 2016). Moreover, expression plasticity (here: the effect that differences of (immune) gene expression converge upon infection) seems to be a general property of sticklebacks, as it has been shown in experiments using sticklebacks from Vancouver Island, Canada (Stutz et al., 2015; Lohman et al., 2017), for sticklebacks from Scotland (Robertson et al., 2015) and for DE sticklebacks infected with tapeworms from across the Northern Hemisphere (Piecyk et al. 2019a). The novelty of this study is that the expression plasticity was studied in

controlled experimental infections, revealing an important effect of the parasite. The immune profiles mostly converged upon infection; but still, Walby and DE expression profiles differed if the fish were infected with NO parasites.

Putting our results in the context of previous studies, host-parasite genotype specific qualitative resistance and gene expression, host genotype specific quantitative resistance and tolerance, as well as converging immune gene expression upon infection are likely a general property of the stickleback-tapeworm system and probably other systems as well. This study moreover presents basic research with important implications for applied clinical sciences. Using helminth infections of sticklebacks, we could, for the first time, show that measures of host immunity (resistance, tolerance, health) need to distinguish between immune response mechanisms along the infection process and need to be put into a co-evolutionary context considering geography and phylogeny as well as environmental variation. The importance of considering the effect of environmental variation on the evolution of immunologically heterogenous populations and infection outcomes is stated more explicitly in the revised manuscript:

“Whether inter-continental resistance can be attributed to local adaptation or specificities of host and parasite populations or clades, warrants further investigation that takes the effect of environmental variation on defence mechanisms and infection outcomes into consideration.” (Lines 313-317)

All information above is included in the manuscript (especially in the Introduction) but, due to word limitations of the journal, cannot be elaborated further.

Referee 1: minor comments, responses and respective changes

1. P7 L140-141. Why isn't host population included as a significant factor ($p = 0.0206$)? I'm guessing it isn't significant after post fdr correction. However, this is not indicated in the legend for Table S1.

Response: *Host population is not included as a significant factor because the significant interaction term implies that significances of main effects are not reliable (i.e., the effect of the main effect relies on the interaction). Fdr correction is not applicable because this analysis does not involve multiple testing.*

2. P7 L146-147: I found this sentence confusing. Perhaps this instead: “Parasite indices (PI), approximations of host quantitative resistance, differed between host populations, but were not affected by parasite strains.

Response: *The sentence stated in lines 146-147 refers to Figure 2, where we structured the data mainly according to host population and show the parasite population in a substructure within the main columns. Thus, the sentence has not been changed in the revised manuscript.*

3. P7 L148: You should mention that DE-Walby contrast for wolf parasites was not significant after fdr correction.

Response: *We do not mention non-significant effects, including effects that are not significant after fdr correction, for three reasons: (i) significances of effects from multiple testing are not reliable without fdr*

corrections, (ii) mentioning all non-significant effects would make the text harder to read and obscure results, and (iii) word limitations of the journal.

4. P9 L177-181: Where are the analysis outputs for these stats? Please add supplementary table number if they are there.

Response: *The corresponding information, including numDF, denDF, test statistics (F) and p-values are provided in the main text:*

“We detected a host-population specific relation between host CF and infection intensity (PI) in Walby and Wolf infections (host population-PI interaction in Walby infections: $F_{2,21.7} = 9.37, p = 0.0012$; host population-PI interaction in Wolf infections: $F_{2,17.5} = 4.02, p = 0.037$) (Figure 3).”

All data and code can also be found at <https://dx.doi.org/10.17617/3.4u>.

5. Supplementary Figures – Many of the supplementary figures (S2, S3, etc) are using wrong/old labels for these populations. Please change for consistency.

Response: *Thank you a lot for pointing this out. All labels of figures and figure legends have been changed for consistency.*

6. Figure S3 – I find it very hard to compare control and treatment values in this figure. Perhaps you could move the controls down with the rest of the data and indicate that the controls are the same for given host population.

Response: *We decided to keep the controls in a separate panel in order to avoid confusion.*

7. P13 L263 – Name the four adaptive genes.

Response: *We named the genes in an earlier version of the manuscript and decided against it in the reviewing process. We are confident that the depth of results in the main manuscript is appropriate and, due to word limitations and in favor of reading flow, decided to mention gene names in the Supplementary Information.*

8. P17 L361-364 – Reword. “*S. solidus* can control stickleback gene expression” is misleading since many parasites can actively control host behavior/phenotype for their own fitness. In this case, I think it would be more apt to say that certain *S. solidus* strains cause convergent gene expression across different stickleback populations (or something to that effect).

Response: *Thank you for pointing this out. In order to acknowledge the fact that Walby- and Wolf-*S. solidus* infections cause convergence, but *NO* does not, the sentence has been changed to:*

“Although the precise molecular mechanisms will have to be investigated, our results suggest that *Schistocephalus solidus* affects stickleback gene expression in a host-parasite genotype dependent manner.” (Lines 356-359)

*On a side note, *S. solidus* is one of the textbook examples for host manipulation. For example, *S. solidus*-infected fish from one of the two Alaskan populations (Wolf) show a distinct white phenotype which has been linked to*

a higher transmission probability to the definitive host. Whether decreased avoidance behaviour (which can be seen across numerous populations) is actively conferred by the parasite or a side effect of energy drain, however, is still a matter of debate.

9. P18 L383 – remove “host” from “hosthost”

Response: *The typo has been corrected.*

10. P18 L385 – Add citation(s)

Response: *The citation for highly diverse infection phenotypes in Alaska has been added and is listed in the References section:*

57. LoBue CP, Bell MA. 1993 Phenotypic Manipulation by the Cestode Parasite *Schistocephalus solidus* of Its Intermediate Host, *Gasterosteus aculeatus*, the Threespine Stickleback. *Am. Nat.* **142**, 725–735.

11. Define parasite sibships and fish families.

Response: *Definitions of ‘parasite sibship’ and ‘fish family’ are included in the revised manuscript:*

“Parasite sibships (offspring from one pair of worms; $n = 4$ per *S. solidus* strain; Figure 1B) were the same in every round; fish families (offspring from one pair of fish) differed between rounds.” (Lines 409-411)

12. P19 L417 – Figure 1 shows 17 individuals per tank. For round 1, which group had one less fish (leading to 16 fish per tank)?

Response: *The tanks from the first experimental round (round 1) housed 3 controls and 13 exposed fish (4 from Walby, 4 from Wolf and 5 from DE). This information is now stated in Figure 1:*

“The table shows sample sizes from experimental rounds 2 and 3; round 1 included 4 exposed fish from Wolf.” (Lines 122-123)

13. Table 2: This table shows the total number of infected stickleback not the proportion.

Response: *You are right - ‘proportion’ has been changed to ‘number’.*

14. P21 L463: Write out what an IFC is.

Response: *The text now includes information on what ‘IFC’ means:*

“Transcription levels of 32 genes were tested using 96.96 Dynamic Array integrated fluidic circuits (IFCs) on a Biomark™ HD system (Fluidigm) with EvaGreen as DNA intercalating dye.” (Lines 459-461)

Referee 2: comments and suggestions with respective changes

1. Lines 368-369: This statement is unclear to me. Please rephrase for clarity!

Response: *This sentence has been rephrased for clarity:*

“We used European and Alaskan three-spined sticklebacks and *Schistocephalus solidus* in experimental infections and found that infection phenotypes were determined by main effects of the host and the parasite.” (Lines 362-364)

2. Line 383: Change ‘hosthost’ to ‘host’

Response: *‘hosthost’ has been changed to ‘host’.*

3. Line 393: Include ‘Max Planck Institute’ to explain the acronym ‘MPI’

Response: *‘Max Planck Institute’ has been included.*

4. Line 428: Replace ‘proportion’ with ‘number’

Response: *‘proportion’ has been changed to ‘number’.*

5. Line 442-446: Explain PI, CF, HSI, SSI and HKI in detail and provide the equations required to calculate these indices! The reader should not have to look this up in the various references.

Response: *Thank you for pointing this out. Information on the various indices were spread across the manuscript and could be found in the Introduction, the Results and the Methods sections. For more clarity, we now state all details in the Methods section. Conceptual information on the parasite index is still included in the Introduction because the features of this index are essential for using this system to study epidemiological traits on different levels and scales.*

“The relative weight of the parasite, the parasite index (PI), was calculated as $100 * \text{parasite weight} / \text{fish weight}$ [35]. Fish condition was estimated by the condition factor (CF, the ratio between observed weight and expected weight at a given length = $100 * \text{fish weight} / \text{fish length}^b$ with fish population-specific exponent b [43] and the hepatosomatic index (HSI) as a measure for medium term energy reserves [44]. The immunological activation was estimated by the splenosomatic index (SSI) and the head kidney index (HKI) [64]. HSI, SSI and HKI were calculated as $100 * \text{organ weight} / \text{fish weight}$. (Lines 439-447)

“The relative weight of *S. solidus* in the fish, the parasite index (PI) [35], is a measure of parasite fitness [36,37], virulence [35,38,39], and host resistance [40].” (Lines 94-96)

6. Line 490: Replace ‘round’ with ‘experimental round’

Response: *This section has been clarified by including the term ‘experimental’ when referring to experimental rounds.*

7. Lines 495-496: “We accounted for the number of copepods that were not ingested” – How was this done?

Response: *We used 2 L aquaria to expose single fish to single copepods: each stickleback was isolated in a 2 L aquarium and starved for one day. We fed the fish with either one uninfected (control) copepod or one singly infected copepod and waited for two days. In order to determine whether the fish ate the copepod, we sieved the water of the single tanks and screened the filters for leftovers. We did not use one filter per tank but pooled the leftovers from each treatment group (i.e., fish family x worm sibship combination). Thus, if we exposed 5 fish and found one copepod in the filter, we corrected the number of exposed fish to ‘four’. This information is stated more clearly in the revised manuscript:*

“Water of each treatment group was sieved and screened for leftover copepods in order to determine the exact number of exposed fish.” (Lines 403-405)

“We determined the infection rate as the proportion of exposed fish (corrected by the number of copepods that have not been eaten) that became infected.” (Lines 437-439)

1. How confident can you be that you would have got the same results if you had used different populations of fish and worms? In other words, are the observed results likely to be a general property of hosts/parasites from different locations, or is it just because you happened to choose these populations? For example, according to lines 381-383, the DE fish have low resistance and the NO worms are highly virulent...would you have got the same results with a highly resistant European fish and a less virulent European worm?

Response: *The presented experiment was designed to study infection outcomes in cross-continental infections. We aimed to investigate the effect of time (along the infection process) and space (geography). We are confident that our findings complement previous work showing that infection success can be linked to geography and, presumably, phylogeny of the host and the parasite (e.g., in Weber et al., 2016; Piecyk et al., 2019a), but that, importantly, environmental variation also is a causal factor in immunological heterogeneity. The fact that the two Alaskan hosts and parasites showed different infection success, even though the lakes are in close proximity, emphasizes the effect of the environment on co-evolutionary trajectories and infection outcomes. In this context, higher baseline immunological activation (as determined for DE fish) has previously been shown for various lake versus river fish and linked to parasite prevalence and abundance (Huang et al., 2016). Moreover, expression plasticity (here: the effect that differences of (immune) gene expression converge upon infection) seems to be a general property of sticklebacks, as it has been shown in experiments using sticklebacks from Vancouver Island, Canada (Stutz et al., 2015; Lohman et al., 2017), for sticklebacks from Scotland (Robertson et al., 2015) and for DE sticklebacks infected with tapeworms from across the Northern Hemisphere (Piecyk et al. 2019a). The novelty of this study is that the expression plasticity was studied in controlled experimental infections, revealing an important effect of the parasite.*

Coming back to your question, we believe that we would get the same results (host-parasite genotype specific qualitative resistance, tolerance and gene expression, host genotype specific quantitative resistance and converging immune gene expression upon infection) with other populations, such as highly resistant European fish and less virulent European worms. Indeed, those effects have already been shown in Piecyk et al., 2019a and Piecyk et al., 2019b. However, more importantly, this study presents a piece of basic research that has important implications for applied clinical sciences. Using this model system, we could, for the first time, show that measures of host immunity (resistance, tolerance, health) need to distinguish between immune response mechanisms and put into a co-evolutionary context involving time and space.

The importance of considering the effect of environmental variation on the evolution of immunologically heterogeneous populations and infection outcomes is stated more explicitly in the revised manuscript:

“Whether inter-continental resistance can be attributed to local adaptation or specificities of host and parasite populations or clades, warrants further investigation that takes the effect of environmental variation on defence mechanisms and infection outcomes into consideration.” (lines 313-317)

2. Lines 381-383: presumably when you say that DE fish have low resistance, you mean to a specific population of worms (presumably a Danish population)? Similarly when saying that NO worms have high growth...or has this been shown to be the case to multiple populations of worms and in multiple populations of fish....? I guess what I'm saying is that you're contending that you've picked a really susceptible European fish and a really virulent European worm, and yet the whole aim of the study is to find out whether this is always the case or whether it depends on what the combination of host and worm is.

Response: *Previous studies have shown that DE stickleback and NO *S. solidus* do represent two extreme forms, thus, two reference points, of hosts and parasites in this model system: DE stickleback are very susceptible to *S. solidus* from various populations (e.g., from Sweden, Germany, Spain, Canada, Iceland, Norway), whereas NO *S. solidus* are highly virulent (as estimated by the parasite index). Unfortunately, due to word limitations, we cannot elaborate on this in the manuscript but provide references for further information:*

“European hosts and parasites are characterised by low resistance against *S. solidus* (DE stickleback) and high growth in sticklebacks (NO *S. solidus*) [31,45,46,56].” (lines 376-378)

Each of the cited publications involves experiments using DE and NO fish and/or parasites.

3. Table 2: the numbers in brackets are supposedly proportions, but proportions should be between 0 and 1: are these percentages? This needs clarifying.

Response: *The table description was misleading. The number in brackets actually shows the number of infected fish. Accordingly, ‘proportion’ has now been changed to ‘number’.*

4. Lines 467-472: how were these genes chosen? I think it important to state this briefly. Off the top of my head and I can see the functional relevance of some of them, but not others, so it might be useful to add a supplementary table stating the function of each gene or the pathways that it is associated with – e.g. you could say that *foxp3* is associated with a regulatory phenotype.

Response: *Thank you for pointing this out. The revised version now includes a supplementary table with all important information on the genes, including gene function, primer sequences, and references:*

“Targets of interest covered four putative reference genes (*b2m*, *ubc*, *rpl13a*, *ef1a* [65]), four regulatory genes (*abtb1*, *ascl1b*, *kat2a*, *mapk13*) and 24 immune genes involved in innate immunity (*marco*, *mst1ra*, *mif*, *il-1 β* , *tnfr1*, *saal1*, *tlr2*, *csf3r*, *p22^{phox}*, *nkef-b*, *sla1*, *cd97*), adaptive immunity (*stat4*, *stat6*, *igm*, *cd83*, *foxp3*, *tgf- β* , *il-16*, *mhcII*, *tcr- β*) and the complement system (*cfb*, *c7*, *c9*) [32,33,46,66] (more information in SI.5).” (lines 580-585)

Literature cited above:

- Huang Y *et al.* 2016 Transcriptome profiling of immune tissues reveals habitat-specific gene expression between lake and river sticklebacks. *Mol. Ecol.* **25**, 943-958. (doi:10.1111/mec.13520)
- Lohman BK, Steinel NC, Weber JN, Bolnick DI. 2017 Gene Expression Contributes to the Recent Evolution of Host Resistance in a Model Host Parasite System. *Front. Immunol.* **8**. (doi:10.3389/fimmu.2017.01071)
- Piecyk A, Roth O, Kalbe M. 2019a Specificity of resistance and geographic patterns of virulence in a vertebrate host-parasite system. *BMC Evol. Biol.* **19**, 80. (doi:10.1186/s12862-019-1406-3)
- Piecyk A, Ritter M, Kalbe M. 2019b. The right response at the right time: Exploring helminth immune modulation in sticklebacks by experimental coinfection. *Mol. Ecol.* **28**, 2668–2680. (doi:10.1111/mec.15106)
- Robertson S, Bradley JE, MacColl ADC. 2015 Measuring the immune system of the three-spined stickleback – investigating natural variation by quantifying immune expression in the laboratory and the wild. *Mol. Ecol. Resour.* **16**, 701-713. (doi:10.1111/1755-0998.12497)
- Scharsack JP, Franke F, Erin NI, Kuske A, Büscher J, Stolz H, et al. 2016 Effects of environmental variation on host–parasite interaction in three-spined sticklebacks (*Gasterosteus aculeatus*). *Zoology* **119**, 375–83.
- Stutz WE, Schmerer M, Coates JL, Bolnick DI. 2015 Among-lake reciprocal transplants induce convergent expression of immune genes in threespine stickleback. *Mol. Ecol.* **24**, 4629–4646. (doi:10.1111/mec.13295)
- Weber JN, Kalbe M, Shim KC, Erin NI, Steinel NC, Ma L, Bolnick DI, Nuismer SL, Michalakis Y. 2016 Resist Globally, Infect Locally: A Transcontinental Test of Adaptation by Stickleback and Their Tapeworm Parasite. *Am. Nat.* **189**, 43-57. (doi:10.1086/689597)